# Similarity thresholds used in DNA sequence assembly from short reads can reduce the comparability of population histories across species

Michael G. Harvey[1,2], Caroline Duffie Judy[1,2,3], Glenn F. Seeholzer[1,2], James M. Maley[1,2,4], Gary R. Graves[3,5] and Robb T. Brumfield[1,2]

[1] Museum of Natural Science, Louisiana State University, Baton Rouge, LA, USA
[2] Department of Biological Sciences, Louisiana State University, Baton Rouge, LA, USA
[3] Department of Vertebrate Zoology, MRC-116, National Museum of Natural History, Smithsonian Institution, Washington, D.C., USA
[4] Moore Laboratory of Zoology, Occidental College, Los Angeles, CA, USA
[5] Center for Macroecology, Evolution and Climate, Natural History Museum of Denmark, University of Copenhagen, Copenhagen Ø, Denmark

Corresponding author
Michael G. Harvey,
mharve9@lsu.edu

## ABSTRACT

Comparing inferences among datasets generated using short read sequencing may provide insight into the concerted impacts of divergence, gene flow and selection across organisms, but comparisons are complicated by biases introduced during dataset assembly. Sequence similarity thresholds allow the *de novo* assembly of short reads into clusters of alleles representing different loci, but the resulting datasets are sensitive to both the similarity threshold used and to the variation naturally present in the organism under study. Thresholds that require high sequence similarity among reads for assembly (stringent thresholds) as well as highly variable species may result in datasets in which divergent alleles are lost or divided into separate loci ('over-splitting'), whereas liberal thresholds increase the risk of paralogous loci being combined into a single locus ('under-splitting'). Comparisons among datasets or species are therefore potentially biased if different similarity thresholds are applied or if the species differ in levels of within-lineage genetic variation. We examine the impact of a range of similarity thresholds on assembly of empirical short read datasets from populations of four different non-model bird lineages (species or species pairs) with different levels of genetic divergence. We find that, in all species, stringent similarity thresholds result in fewer alleles per locus than more liberal thresholds, which appears to be the result of high levels of over-splitting. The frequency of putative under-splitting, conversely, is low at all thresholds. Inferred genetic distances between individuals, gene tree depths, and estimates of the ancestral mutation-scaled effective population size ($\theta$) differ depending upon the similarity threshold applied. Relative differences in inferences across species differ even when the same threshold is applied, but may be dramatically different when datasets assembled under different thresholds are compared. These differences not only complicate comparisons across species, but also preclude the application of standard mutation rates for parameter calibration. We suggest some best practices for assembling short read data to maximize comparability, such as using more liberal thresholds and examining the impact of different thresholds on each dataset.

## INTRODUCTION

With the proliferation of population-level datasets obtained using massively parallel sequencing technologies, there is increasing interest in studies that compare inferences from genomic datasets obtained from different species (e.g., *Leaché et al., 2013*; *Smith et al., 2013*) or from different genomic regions (e.g., *Evans et al., 2014*; *Harvey et al., 2013*; *Leaché et al., 2015*). Assembly of short sequence reads into orthologous loci is a key component of post-sequence processing, and commonly used methods can lead to biases in population genetic parameter estimation (*Ilut, Nydam & Hare, 2014*). Here, we explore the effect of one major source of bias on the comparability of datasets and inferences.

Sequence similarity provides the information necessary for assembling reads into orthologous loci (*Pop & Salzberg, 2008*; *Chaisson, Brinza & Pevzner, 2009*). By setting a sequence similarity threshold, researchers attempt to assemble similar, presumably orthologous reads into loci while separating or removing dissimilar, presumably non-orthologous reads (e.g., *Etter et al., 2011*; *Catchen et al., 2011*). Selecting the most appropriate similarity threshold is challenging, primarily because the amount of genetic (allelic) variation can vary greatly among orthologous loci within a species (*Ilut, Nydam & Hare, 2014*). Because the amount of genetic variation also varies among species and genomic regions, a particular similarity threshold may impact each dataset differently, potentially influencing inferences in comparative studies.

Many methods default to a stringent similarity threshold, often requiring 98–99% sequence similarity among reads for assembly (e.g., *Catchen et al., 2011*; *Lu et al., 2013*). However, stringent similarity thresholds may split orthologous reads into multiple loci if the reads come from alleles that are more different than the threshold permits (hereafter "over-splitting"; Fig. 1A). More liberal similarity thresholds permit the assembly of more dissimilar orthologous reads into loci, but are more susceptible to including paralogous reads in the assembly (hereafter "under-splitting"; Fig. 1B). Using simulations, *Rubin, Ree & Moreau (2012)* found that under-splitting was frequent at more liberal similarity thresholds in phylogenetic datasets, but did not strongly bias inference. *Catchen et al. (2013)* examined RAD-Seq data from three-spined sticklebacks, and found that over-splitting was an issue when datasets were processed with similarity thresholds more stringent than 96%. *Ilut, Nydam & Hare (2014)* tested the impact of similarity threshold selection on both over- and under-splitting in three simulated and one empirical RAD-Seq dataset. They found that under-splitting was minimal and that affected loci were easily identified due to the presence of individuals with more alleles than expected given their ploidy, but that over-splitting was significant at more stringent similarity thresholds.

Comparative phylogeographic and population genetics studies are particularly susceptible to biases resulting from similarity thresholds, particularly over-splitting. Different species often exhibit different levels of genetic diversity (*Lewontin, 1974*; *Taberlet*

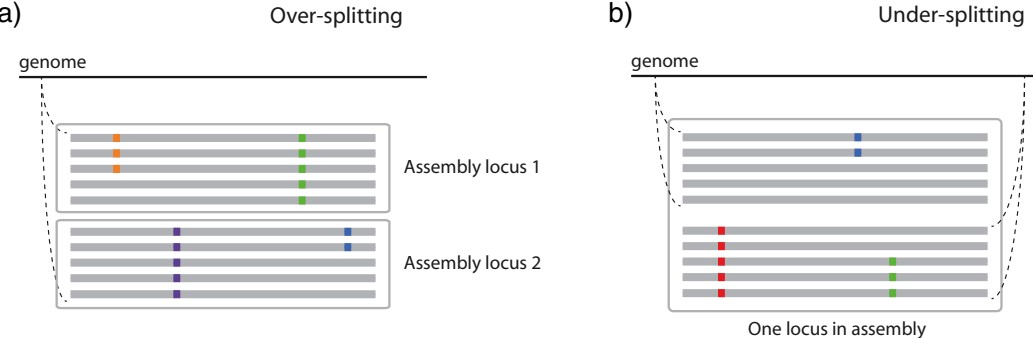

**Figure 1  Two ways in which similarity thresholds can result in spurious assemblies.** (A) over-splitting occurs when reads from different alleles from the same genomic position are spuriously split into multiple loci due to lower similarity than the similarity threshold parameter, and (B) under-splitting occurs when reads from different genomic positions are clustered into a single locus due to higher similarity than the similarity threshold parameter. Gray bars represent identical sequence across reads, whereas colored squares represent alternate alleles at SNPs.

*et al., 1998*; *Smith et al., 2014*; *Romiguier et al., 2014*), and this variation across species may interact with the application of similarity thresholds to differentially bias datasets. *Huang & Knowles (in press)*, for example, found that mutational spectra of datasets simulated under deeper species trees were biased downwardly relative to those simulated under shallow species trees when processed with the same similarity threshold (both 98% and 95% similarity were examined). The effects of similarity thresholds have not been examined, however, using empirical data from species that vary in their levels of genetic diversity. Although diverse parameters required for short read assembly are worthy of scrutiny, we focus on similarity thresholds as they are particularly important for maintaining comparability across species with different levels of variation.

In this study, we examine the effect of similarity thresholds on dataset assembly and phylogeographic inferences across four non-model bird lineages that vary in divergence. We sample two populations or species within each lineage and assemble a RAD-Seq dataset for each lineage at a series of similarity thresholds to assess the impact of different thresholds on the number of unique alleles observed within assembled loci. We investigate the effect of different similarity thresholds on estimates of standard population genetic and phylogeographic parameters within species and in comparisons across species.

## MATERIALS AND METHODS

### Study species and sampling

We sampled four individuals from each of two populations, as determined based on taxonomy and prior genetic data, in four lineages (Table S1). The first lineage includes Clapper (*Rallus crepitans* JF Gmelin, 1788) and King (*R. elegans* JJ Audubon, 1834) rails, sister species of medium-sized water birds that interbreed in a narrow hybrid zone centered on a salinity gradient (*Maley, 2012*; *Maley & Brumfield, 2013*). We also examined the Streamertail (*Trochilus polytmus* C Linnaeus, 1758), a hummingbird endemic to Jamaica that comprises two subspecies (*T. p. polytmus* and *T. p. scitulus*) that differ primarily in
bill coloration and interbreed in a narrow hybrid zone (*Gill & Stokes, 1973*; *Coyne & Price, 2000*). Line-cheeked (*Cranioleuca antisiensis* PL Sclater, 1859) and Baron's (*Cranioleuca baroni* O Salvin, 1895) Spinetails are closely related, small insectivorous birds distributed along the Andes Mountains (*Remsen, 2003*). Finally, we sampled two populations of Plain Xenops (*Xenops minutus* AE Sparrman, 1788), a widespread insectivorous bird of lowland Neotropical forests that are separated by the Andes and differ in plumage, voice, and genetic markers (*Remsen, 2003*; *Burney, 2009*; *Harvey & Brumfield, 2015*).

## Laboratory methods

For each individual examined, we extracted total DNA from vouchered tissue samples using DNeasy tissue kits (Qiagen, Valencia, California, USA) following the manufacturer's protocol. We sent DNA extracts to the Cornell Institute of Genomic Diversity (IGD) to collect data using Genotyping by Sequencing, a RAD-Seq method (*Elshire et al., 2011*). Briefly, the IGD digested DNA using PstI (CTGCAG) and ligated a sample-specific indexed adapter and common adapter to resulting fragments. The IGD pooled and cleaned ligated samples using a QIAquick PCR purification kit (Qiagen, Valencia, CA, USA), amplified the pool using an 18-cycle PCR, purified the PCR product using QIAquick columns, and quantified the amplified libraries using a PicoGreen assay (Molecular Probes, Carlsbad, California, USA). Based on the PicoGreen concentrations, the IGD then combined the samples for this project with unrelated samples and ran plates of 96 samples on a 100-base pair, single-end Illumina HiSeq 2000 lane (Illumina, San Diego, California, USA).

## Bioinformatics processing

We processed the raw GBS reads using the Stacks pipeline (*Catchen et al., 2011*; *Catchen et al., 2013*) due to its popularity in prior studies assembling RAD-Seq datasets within species. Although other dataset assembly programs are available (e.g., *Eaton, 2014*; *Sovic, Fries & Lisle Gibbs, in press*), all rely on similarity thresholds and should yield similar results with respect to the analyses presented here. Datasets were assembled on compute nodes (2.93 GHz Quad Core Nehalem Xeon 64-bt processors with 24 GB 1,333 MHz RAM or 96 GB 1,066 MHz RAM) maintained by LSU High Performance Computing. We demultiplexed raw reads, cleaned reads, and removed barcode and adapter sequences using the program process_radtags.pl. We assembled alleles and loci *de novo* using the program denovo_map.pl. We used custom Python (*Python Software Foundation, 2007*) scripts (available at https://github.com/mgharvey/misc_Python) to obtain sequence alignments of both alleles for each individual from the Stacks output files. Detailed settings are provided in the supplement (Table S2).

To investigate the impact of similarity thresholds on dataset attributes and downstream analyses, we assembled seven datasets for each of the four lineages under similarity thresholds (Stacks settings -M and -n) at all integer values from 93% (7 mismatches allowed) to 99% (1 mismatch allowed), reflecting the range of settings typically used for assembling intraspecific datasets. Assembly with similarity thresholds less stringent than 93% failed due to high computational demand in Stacks, but should not be necessary for the divergences examined here or for most other population-level studies. Reads

with similarity values above the selected threshold clustered into assemblies, which we treated as independently segregating loci in downstream analyses. We disabled the use of secondary, more divergent reads for calling genotypes (Stacks setting -H) to prevent the assembly of reads that are less similar than the similarity threshold used for primary stacks. We set minimum depth per allele (Stacks setting -m) to ten, which provides a balance between the inclusion of singleton alleles (potential errors) and the total size of the data matrix (Fig. S1). We set the maximum number of alleles per individual (Stacks setting –max_locus_stacks) to three, one above the ploidy level of the study organisms. In the resulting datasets, this setting will result in three called alleles for any individuals containing three or more alleles, allowing the identification of alignments containing reads from paralogous loci. We used custom Python scripts to format files and calculate basic statistics and used COMPUTE (*Thornton, 2003*) to estimate standard population genetic summary statistics. Monomorphic loci as well as those with variable sites were retained in all subsequent analyses unless otherwise specified.

## Number of alleles

We examined the number of unique alleles per locus across treatments to determine how different similarity thresholds affected each dataset. As an index of the frequency of under-splitting in each dataset, we calculated the number of loci containing individuals with more than two alleles. These loci were presumed to contain paralogous reads and were removed from further analysis. To assess the proportion of loci with putative over-split alleles, we mapped loci assembled under the more stringent thresholds (94–99%) to the set of loci assembled under the most liberal threshold (93%). This allowed us to detect instances in which multiple loci from the more stringent threshold mapped to the same locus from the liberal threshold. We used LASTZ (*Harris, 2007*) for mapping with minimum identity set at 93% for all comparisons and no gaps permitted. We subtracted from each total the number of loci from the liberal threshold (93%) that mapped to other loci assembled with the same threshold using LASTZ.

## Genetic distances and $F_{st}$

Over-splitting may reduce estimates of genetic distance between individuals or populations if they contain dissimilar alleles. Conversely, if over-splitting reduces the number of alleles within populations, this may reduce estimates of distance between populations. We calculated pairwise *p*-distances and Jukes-Cantor corrected distances per unit sequence length at each locus. We measured distances between individuals by measuring the average distance between alleles in the first individual and those in the second individual. For loci containing variable sites, we also estimated $F_{st}$ between the two populations within each lineage using formula (3) of *Hudson, Slatkin & Maddison (1992)*, which is based on the ratio of the mean number of differences between different sequences sampled within populations to the mean number of differences between sequences sampled between populations.

## Gene trees

Over-splitting may also reduce average gene tree depth due to the loss of more variable loci owing to them being subdivided into two or more less variable loci. To reduce computation, we selected a random subsample of 1,000 loci for each lineage at each threshold for gene tree estimation. We selected the best-fit finite sites substitution model for each locus using mrAIC.pl (*Nylander, 2004*) and conducted MrBayes (*Ronquist & Huelsenbeck, 2003*) runs with a random starting tree, four Markov chains, and a 100,000-iteration burn-in followed by 1,000,000 sampling iterations. We measured the depth of gene trees as mean depth of the deepest node in number of expected substitutions using the R (*R Core Team, 2014*) package ape (*Paradis, Claude & Strimmer, 2004*).

## Demographic parameter estimation

We used the 1,000 locus subsets from gene tree estimation to estimate ancestral and contemporary population sizes in each lineage at each similarity threshold using the coalescent model implemented in BP&P (*Yang & Rannala, 2010*). Although this method assumes no gene flow between populations, which may be violated in some of our study lineages, simulations have demonstrated that BP&P performance is robust to limited gene flow (*Zhang et al., 2011*). We used a speciation model containing two contemporary populations and a divergence time parameter ($\tau$) as well as population standardized mutation rate parameters ($\theta = 4N_e\mu$, where $N_e$ is the effective population size and $\mu$ is the substitution rate per site per generation) for both daughter populations and an ancestral population. We set prior values using gamma distributions determined by a shape parameter ($\alpha$) and scale parameter ($\beta$). Priors for both divergence time and population standardized mutation rate were set to $\alpha = 1$ and $\beta = 300$. We ran analyses for a burn-in of 50,000 iterations and then sampled every other iteration for an additional 500,000 iterations.

## RESULTS

After removing loci containing putative paralogous reads (see below), we recovered between 96,776 and 158,328 loci for the four lineages across the range of similarity thresholds examined (Table 1). The similarity threshold used had an effect on the number of unique alleles per locus in all four lineages (Kruskal Wallis test $p < 2.20^{-16}$; Table S3). The number of alleles was low using the 99% similarity threshold, but increased and plateaued as the threshold approached 93% (Fig. 2A). The number of alleles was more similar across lineages at stringent thresholds relative to liberal thresholds. For example, *Xenops* contained, on average, 1.4 times as many alleles as *Rallus* when processed with a 99% similarity threshold, but 1.66 times as many alleles when processed with a 93% similarity threshold.

The proportion of loci containing putative paralogous reads (under-split loci) increased slightly with increasing similarity thresholds, but was less than 0.4% at all thresholds for all lineages (Fig. 2B). At all thresholds, *Trochilus* exhibited roughly half the level of putative paralogy displayed in the other lineages (Table S4). Depending on the lineage, 5–61% of loci represented putative over-split alleles based on LASTZ mapping at the most stringent
**Table 1** Attributes and summary statistics (standard deviation across loci) of datasets assembled under the similarity thresholds examined.

| | Threshold | Loci | Individuals represented per locus | Segregating sites per locus |
|---|---|---|---|---|
| | 99 | 147,123 | 4.14 (1.82) | 0.20 (0.44) |
| | 98 | 145,423 | 4.2 (1.82) | 0.30 (0.63) |
| | 97 | 144,475 | 4.21 (1.82) | 0.34 (0.74) |
| *Cranioleuca* | 96 | 143,780 | 4.22 (1.82) | 0.38 (0.86) |
| | 95 | 142,897 | 4.23 (1.82) | 0.41 (0.98) |
| | 94 | 141,880 | 4.23 (1.82) | 0.44 (1.11) |
| | 93 | 140,801 | 4.24 (1.81) | 0.48 (1.26) |
| | 99 | 100,086 | 3.3 (1.31) | 0.17 (0.41) |
| | 98 | 99,300 | 3.31 (1.31) | 0.24 (0.60) |
| | 97 | 98,680 | 3.31 (1.31) | 0.28 (0.73) |
| *Rallus* | 96 | 98,206 | 3.31 (1.3) | 0.30 (0.83) |
| | 95 | 97,808 | 3.31 (1.3) | 0.33 (0.93) |
| | 94 | 97,321 | 3.31 (1.3) | 0.36 (1.07) |
| | 93 | 96,776 | 3.32 (1.3) | 0.40 (1.22) |
| | 99 | 125,594 | 3.83 (1.67) | 0.32 (0.56) |
| | 98 | 125,966 | 3.87 (1.7) | 0.46 (0.77) |
| | 97 | 125,697 | 3.88 (1.7) | 0.51 (0.87) |
| *Trochilus* | 96 | 125,437 | 3.88 (1.7) | 0.54 (0.95) |
| | 95 | 125,118 | 3.89 (1.7) | 0.56 (1.02) |
| | 94 | 124,669 | 3.89 (1.7) | 0.59 (1.13) |
| | 93 | 123,926 | 3.9 (1.7) | 0.62 (1.25) |
| | 99 | 155,933 | 3.77 (1.71) | 0.65 (0.79) |
| | 98 | 158,496 | 3.94 (1.74) | 1.05 (1.17) |
| | 97 | 158,281 | 4 (1.74) | 1.25 (1.41) |
| *Xenops* | 96 | 158,328 | 4.01 (1.74) | 1.35 (1.56) |
| | 95 | 158,078 | 4.02 (1.74) | 1.40 (1.66) |
| | 94 | 157,534 | 4.02 (1.74) | 1.45 (1.76) |
| | 93 | 156,640 | 4.03 (1.74) | 1.50 (1.87) |

similarity threshold of 99%, but putative over-split alleles decreased as thresholds became more liberal (Fig. 2B).

Both uncorrected $p$ and Jukes-Cantor corrected genetic distances between individuals were reduced at more stringent similarity thresholds (Fig. 3A). Variance across lineages in mean genetic distance increased as similarity thresholds became more liberal (Fig. S2), although relative values between lineages were similar across thresholds. $F_{st}$ estimates between populations did not differ across thresholds (Fig. 3B).

Mean gene tree depth, based on the depth of the deepest node, increased as more liberal similarity thresholds were applied in each lineage (Fig. 3C). Variance in mean gene tree depths across lineages was inversely related to threshold stringency (Fig. S2) and relative values across lineages were contingent on the threshold applied. For example, the

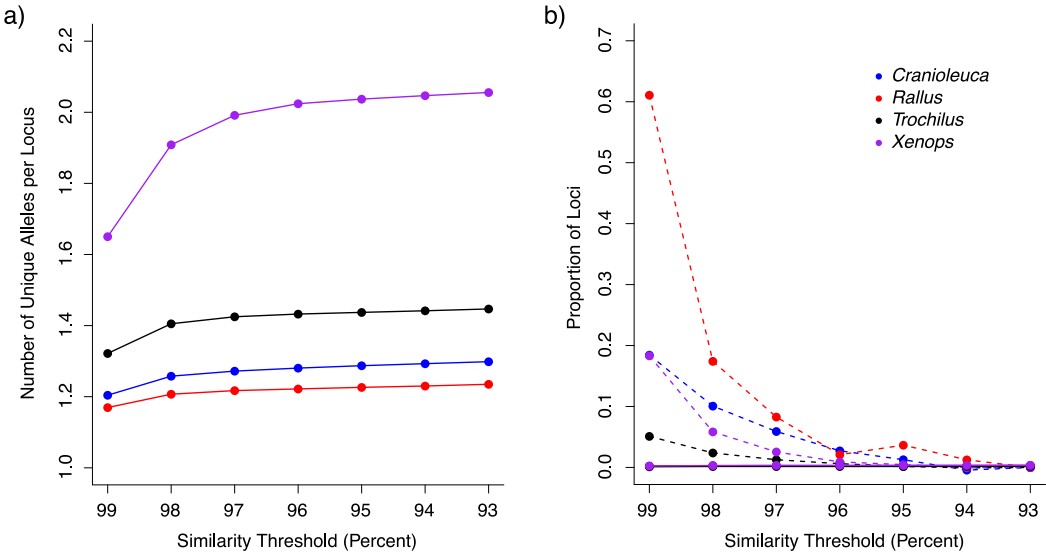

**Figure 2 The impact of similarity thresholds on empirical datasets from four bird lineages.** (A) Stringent similarity thresholds resulted in fewer unique alleles per locus relative to more liberal thresholds. (B) Putative over-split loci (connected by dashed lines) were more frequent in datasets assembled at stringent similarity thresholds, whereas loci containing under-split reads (solid lines) occurred at low frequency across all similarity thresholds (lines are overlapping).

mean gene tree depth for *Xenops* was $1.48\times$ greater than for *Rallus* at 99% similarity, but $1.91\times$ greater at 93% similarity.

Ancestral $\theta$ estimates were higher at more liberal similarity thresholds for all four lineages (Fig. 3D), but contemporary $\theta$ estimates and population divergence times ($\tau$) showed no association with similarity thresholds (Figs. S3 and S4). Ancestral $\theta$ estimates, as with genetic distance and gene tree depth, displayed lower variance across lineages at stringent relative to liberal thresholds (Fig. S2). Relative values across lineages also differed across thresholds. The ancestral $\theta$ for *Xenops* was $1.89\times$ greater than for *Rallus* at 99% similarity, for example, but $2.95\times$ greater at 93% similarity.

## DISCUSSION

Comparability of parameter estimates is essential for comparative studies of phylogeographic structure and genetic diversity across species or among genomic regions (*Nybom, 2004*). Our results reveal, however, that inferences differ not only among lineages with different population histories, but also according to the similarity threshold applied during dataset assembly. Differences in the impact of similarity thresholds across datasets not only reduce the utility of those datasets for comparative studies, but also preclude the application of standardized mutation rate estimates that would allow demographic parameters in non-model species to be converted to absolute values (*DaCosta & Sorenson, 2014*). The issues discussed here are not restricted to RAD-Seq datasets, but are of concern for all short read datasets requiring similarity-based *de novo* assembly, including those from sequence capture and transcriptomic approaches. Mapping reads to existing reference sequences also requires the application of similarity thresholds and, although

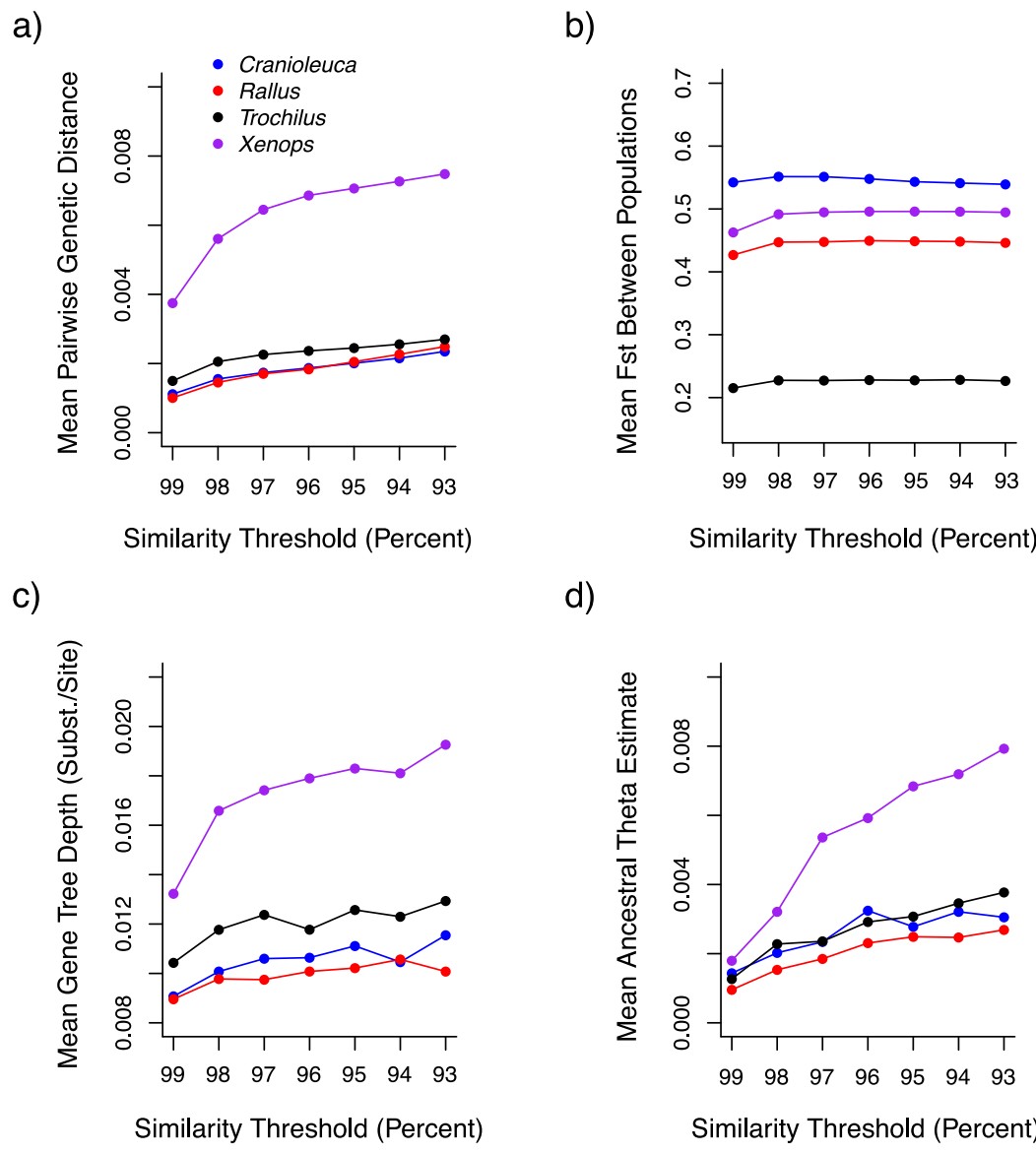

**Figure 3 The impact of similarity thresholds on population genetic parameter estimates.** The similarity threshold applied impacts (A) mean pairwise Jukes-Cantor corrected genetic distance between individuals, (B) mean $F_{ST}$ between populations, (C) mean gene tree depth and (D) ancestral theta ($\theta$) based on a coalescent model.

identifying under-splitting is more straightforward with a reference genome, divergent alleles may still be lost if the threshold used for mapping is too stringent (*Trapnell & Salzberg, 2009*; *Lunter & Goodson, 2011*). In such cases, over-splitting results in the loss of alleles divergent from the reference, rather than the splitting of alleles into separate loci. Careful selection of similarity thresholds for assembly is an important issue for diverse sequencing projects, particularly if comparisons are to be made across datasets.

We found that datasets assembled under stringent similarity thresholds included fewer unique alleles per locus than those assembled under more liberal thresholds. Similarly, *Ilut, Nydam & Hare (2014)* found heterozygosity was reduced when stringent similarity

thresholds were applied, but increased with more liberal thresholds across three simulated and one empirical dataset. The reduced number of alleles per locus in datasets assembled with stringent thresholds is likely due to the higher frequency of putative over-splitting in those datasets. Prior studies also demonstrated that over-splitting is frequent when datasets are processed at stringent similarity thresholds, and that this leads to allele loss (*Catchen et al., 2013*; *Ilut, Nydam & Hare, 2014*). Our results suggest that under-splitting occurs at low frequencies across similarity thresholds and has little impact on datasets. The impact of under-splitting may be more severe in species with highly repetitive genomes or in studies across deep phylogenetic timescales that require more liberal similarity thresholds for assembly (e.g., *Rubin, Ree & Moreau, 2012*; *Eaton & Ree, 2013*).

Variation in datasets resulting from the similarity threshold applied has important effects on downstream parameter estimation. In addition to the biases in population genetic and phylogeographic estimates that we found, *Huang & Knowles (in press)* found that mutational spectra are downward-biased as a result of the loss of the most divergent alleles, and some studies have found that phylogenetic estimates are more accurate when more liberal similarity thresholds are applied to simulated data (*Rubin, Ree & Moreau, 2012*; *Huang & Knowles, in press*). Unlike other parameters, our $F_{st}$ estimates were not strongly impacted by variation in similarity thresholds, perhaps because $F_{st}$ is calculated using the ratio of between- and within-population divergence, both of which are impacted by allele loss. In addition, $\theta$ values from contemporary populations were similar across thresholds, while ancestral $\theta$ values were lower at more stringent thresholds. This may result if stringent thresholds result in the loss of alleles that are fixed between the two divergent populations at a higher rate than those that are variable within populations. Despite these exceptions, it seems likely that observed biases in datasets across similarity thresholds would impact diverse population genetic and phylogeographic parameter estimates.

Stringent similarity thresholds (98–99%) are widely applied currently to population-level studies (e.g., *Emerson et al., 2010*; *Reitzel et al., 2013*; *Chu et al., 2014*), perhaps under the supposition that they are more conservative and less likely to permit the assembly of non-orthologous reads or as an attempt to reduce dataset size and computation times (*Ilut, Nydam & Hare, 2014*). We concur with *Ilut, Nydam & Hare (2014)* and *Huang & Knowles (in press)* that defaulting to stringent thresholds is generally not appropriate. Over-splitting decreases at more liberal similarity thresholds and the number of alleles per locus asymptotes near the 96% threshold, suggesting that datasets assembled under similarity thresholds of 96% or less stringency are relatively less biased by over-splitting. Although this asymptote will vary depending on the divergence within a dataset, other studies have found asymptotes at similar threshold values, for example at roughly 95–96% in empirical data from sticklebacks (*Catchen et al., 2013*) or between roughly 88% and 96% in simulated tunicate, stickleback, and soybean datasets and an empirical tunicate dataset (*Ilut, Nydam & Hare, 2014*). The approach suggested by *Ilut, Nydam & Hare (2014)* in which datasets are assembled at a series of similarity thresholds, the location of the asymptote in over-splitting is identified, and that threshold is used for final assembly is preferable to defaulting to stringent thresholds.

We were unable to directly investigate the frequency of under-splitting and over-splitting in our datasets because we lack genome sequences for the non-model organisms examined. Our indirect measure of over-splitting may detect not just over-split loci, but also loci that are under-split in the assembly from the most liberal threshold but correctly separated in the assembly from the more stringent thresholds. This would be particularly likely if paralogy was common in the genomes under investigation or if very liberal similarity thresholds were examined. The frequency of under-splitting appears to be low enough in our datasets, however, that this effect would be minimal. Broad concordance between our results and prior investigations into over-splitting in systems with a genome for reference (*Catchen et al., 2013*; *Ilut, Nydam & Hare, 2014*) suggest that our metric of over-split alleles is a reasonable proxy for use in non-model organisms.

Results from our indirect measure of under-splitting are also broadly consistent with the low levels of under-splitting observed in prior work using reference genomes (*Ilut, Nydam & Hare, 2014*) and were expected given the low level of paralogy in avian genomes (e.g., chicken; *Hillier et al., 2004*). Our measure of under-splitting, the number of loci containing individuals with more alleles than expected, has been used previously to filter out loci with paralogous data from RAD-Seq datasets (*Parchman et al., 2012*; *Peterson et al., 2012*). Some loci may contain reads from paralogous loci but may not contain sufficient numbers of alleles to trip this filter, potentially inflating estimates of variation. Prior work, however, suggests that paralogous reads lack strong signal conflicting with that from entirely orthologous loci and have relatively minor effects on inferences (*Rubin, Ree & Moreau, 2012*). Other indicators such as extreme heterozygosity or other deviations from Hardy–Weinberg or linkage equilibrium in presumed panmictic populations (*Catchen et al., 2011*; *White et al., 2013*), violations of Mendelian inheritance in pedigreed individuals, or gene tree topologies suggesting a history of duplication might also be used to detect additional loci containing paralogous reads. These metrics deserve consideration in situations where under-splitting is a concern, including in species with high levels of heterozygosity or deep divergences (e.g., phylogenetic studies) necessitating the application of very liberal similarity thresholds or in species with highly repetitive genomes.

We uncovered differences in allelic diversity and parameter estimates across the four study lineages examined. *Xenops minutus* generally displayed the greatest allelic diversity and also the largest genetic distances between individuals, deepest gene trees, and highest $\theta$ values, which was perhaps not surprising given prior evidence of deep genetic divergences within this species (*Smith et al., 2014*; *Harvey & Brumfield, 2015*). The other lineages were more similar by most measures, although *Trochilus polytmus* was slightly higher than *Cranioleuca* and *Rallus* in allelic diversity, genetic distance, and gene tree depths. Interestingly, *Trochilus polytmus* also exhibited roughly half the frequency of putative paralogous loci of the other three species, which may be related to the small genome size of hummingbirds (*Gregory et al., 2009*).

Our results suggest that the similarity threshold used for assembly impacts the level of variation in a dataset as well as downstream population genetic and phylogeographic estimates. Comparisons across datasets are also biased by the impact of similarity

thresholds, appearing more similar across datasets when stringent thresholds are used or in some cases more different if species are assembled with different thresholds. These biases further preclude the estimation of standardized mutation rates for parameter calibration. Methods for threshold selection exist that limit these biases, such as the use of liberal thresholds and examination of the impact of a range of thresholds on a given dataset, but they need to be further developed and applied more widely if we are to be able to compare datasets and integrate inferences across studies, genomic regions, and organisms.

## ACKNOWLEDGEMENTS

We thank the many collectors and museum curators and staff involved in obtaining and maintaining the samples used for this study, in particular John M. Bates and David E. Willard (FMNH), Mark B. Robbins (KUMNH), Donna L. Dittmann (LSUMNS), and James Dean (USNM). LSU High Performance Computing provided computer resources. Jeremy M. Brown, Michael E. Hellberg, Prosanta Chakrabarty, Jacob A. Esselstyn, and the LSU Vert Lunch group provided helpful comments on the issue addressed in this paper. We thank Scott V. Edwards, Ryan C. Garrick, Magnus Popp, Paul Hohenlohe, and three anonymous reviewers for constructive comments that improved the manuscript.

### Funding

Funding was provided by NSF Doctoral Dissertation Improvement Grants to Michael G. Harvey (DEB-1210556) and James M. Maley (DEB-1110624). The funders had no role in study design, data collection and analysis, decision to publish, or preparation of the manuscript.

### Grant Disclosures

The following grant information was disclosed by the authors:
NSF Doctoral Dissertation Improvement: DEB-1210556, DEB-1110624.

### Competing Interests

Caroline Duffie Judy and Gary R. Graves are employees of The National Museum of Natural History, Smithsonian Institution.

### Author Contributions

- Michael G. Harvey conceived and designed the experiments, performed the experiments, analyzed the data, contributed reagents/materials/analysis tools, wrote the paper, prepared figures and/or tables, reviewed drafts of the paper.
- Caroline Duffie Judy, Glenn F. Seeholzer, James M. Maley, Gary R. Graves and Robb T. Brumfield conceived and designed the experiments, contributed reagents/materials/analysis tools, wrote the paper, reviewed drafts of the paper.

## DNA Deposition

The following information was supplied regarding the deposition of DNA sequences:
    NCBI Sequence Read Archive (Project: PRJNA280209).

## Data Deposition

The following information was supplied regarding the deposition of related data:
    Misc_Python (doi: 10.5281/zenodo.15693).

## Supplemental Information

Supplemental information for this article can be found online at http://dx.doi.org/10.7717/peerj.895#supplemental-information.

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
