# Peer review of "Similarity thresholds used in DNA sequence assembly from short reads can reduce the comparability of population histories across species"

_PeerJ, doi:10.7717/peerj.895_

## Round 0.1 · original submission · Minor Revisions

Dear Michael -

Your manuscript has been reviewed by two experts in the field and I am delighted to report that both of them found only minor issues that need to be addressed. These mainly involve clarifications of various phrases. I will weigh in on one comment by Reviewer 1, who suggests for line 84 using the term "downwardly-biased". Rather I would suggest not using the hyphen, but retaining "downwardly biased" or even better "biased downwardly".

I might suggest one additional comment, in the title: as written it sounds as if implementing any threshold will reduce comparability, but what I think you mean is using different thresholds will do this. Consider either using "Different" in the title or using something like "can" to indicate that sometimes comparability will be reduced, but not always.

Otherwise I think the comments should be fairly easy to address and I look forward to seeing your revisions.

·

Basic reporting

Good. See detailed comments (below) for suggested improvements.

Experimental design

Good. No issues here.

Validity of the findings

Good. No issues here..

Additional comments

TITLE

Suggestion: add words “DNA sequence” (or similar).


ABSTRACT

A little vague: “Comparing inferences among datasets generated using short read sequencing may provide insight into the concerted effects of evolutionary processes across organisms...” What kinds of inferences (e.g., demographic, structure, gene flow, divergence)?

Consider expanding: “...assembly of short reads into [pools of genetically similar allele haplotypes representing different] loci...”(or similar)

Clarification: define “stringent thresholds” as meaning those that require high genetic similarity (or similar)

Add words: “...with different levels of [within-lineage] genetic divergence...”

Delete words: “...high levels of over-splitting [at stringent thresholds]"


INTRODUCTION

L84: downwardly-biased (rather than downward-biased)?

L85-86: Clarification: is it correct that the similarity threshold used by Huang & Knowles was stringent? If so, add that information here. Also, it might be worth expanding what is meant by “when processed with the same settings” (I think this refers to criteria for excluding loci based on the proportion of individuals with missing genotypes, but not sure if this is what you mean).

L93 (and elsewhere): I presume that “number of alleles observed” refers to number of non-redundant (i.e., unique) alleles, rather than the number of allele copies sampled. However, it may be worth clarifying this.


METHODS

L103, Clarification: on what basis do you define a “population”? (e.g., geography, prior genetic analyses, etc).

L109-112, Inconsistency: the text here says “The Line-cheeked Spinetail (Cranioleuca antisiensis P. L. Sclater, 1859)... from which we sampled two subspecies (C. a. antisiensis and C. a. baroni)...”. However in Table S1, the named antisiensis and baroni are applied at the species (not subspecies) level.

L112-113, Comment: whatever the definition of population (above), I think that it is being overly loosely applied in this case. Here it is stated that two populations of Xenops minutus were sampled, yet Table S1 shows that representative of three subspecies were included, with X. m. mexicanus and X. m. littoralis being considered part of the same population. It is not immediately obvious to me how a single population can comprise members of more than one subspecies.

L136: Spelling / clarification: artefacts (rather than artifacts)? Also, I think it would be good to specify what kind of artefacts you refer here. For example, are you referring only to stringency thresholds for clustering of alleles into groups that are each considered to be a different locus?

L142: Clarification: “Detailed settings are provided in the supplement”. Which part of the supplementary material are you referring to here... is it Fig. S1?

L143: impact of similarly thresholds on what? It’s probably best to be explicit about this.

L155: Meaning unclear: If the maximum number of alleles per individual was set at three (i.e., diploid +1), how is it possible to “...detect loci containing individuals with three _or more_ alleles”?

L162 Add words: “... number of [unique] alleles per locus...” (or similar)

L164: Does “more than two alleles” mean three alleles here? As above, if the maximum number of alleles per individual was set a three, then it would be good to just clearly say three here.

L169, Missing word: “... the same locus from the [less] stringent threshold” (is this correct?)

L170-172: Just an idea here: The difference between that number of loci obtained using the most liberal (93% similarity) threshold from a more stringent one (i.e., 92-99%) does really give much insight into the manner in which loci had potentially been over-split in the latter cases. For example, is it possible to determine whether the most common situation is that of two loci collapsing into one (or three into one, or four into one), and by extension, whether this is constant (or not) over the different types of stringency thresholds (i.e., 92-99% vs. 93%)?

L176, Give rationale: “Over-splitting may reduce estimates of genetic distance between... populations”. In the context of Weir & Cockerham's (1984) estimate of Fst, my initial reaction is the opposite. The reason is as follows: the maximum Fst value (Fst = 1) is attainable only when each population is fixed for a different allele, and in general, the more alleles at a locus, the more depressed the largest possible Fst value becomes (e.g., microsat loci almost always produce very small Fst values). So, given that over-splitting of loci will tend to reduce allelic variance within populations, it should also tend to increase that maximum possible Fst. This should be true when Fst is calculated in a way that accounts only for allele frequency information (e.g., Weir & Cockerham; but not when divergence among alleles is taken into account, as in Rst or analogs).

You refer to using Hudson et al’s estimate of Fst, which is based on estimating Fst from frequencies of polymorphic sites, treating each site as a separate locus (side note: I think this information should be added to the text). In that paper, Hudson et al. state that their estimate (using eq. 3) is numerically identical to Weir & Cockerham's when sample sizes from each population are equal, which I assume is true in the present paper. Also, in your Fig. 3b, there is a (very weak) indication that under-splitting (99% threshold) depresses Fst (cf. other thresholds)... which is somewhat consistent with the rationale that I’ve given. So... I just wanted to flag that statement of L176 (as it relates to Fst) as potentially being counter-intuitive, and so should be expanded of rephrased.

L178-179, Clarification: does “per unit sequence length” refer to a ~100bp alignment? Also, at the level of pairwise distances between individuals, is it true that for each pair of individuals, distances (p, and JC) were an average of n = 4 comparisons (i.e., given 2 alleles per diploid genotype, and two genotypes to be compared)? This description is just a little too vague for me to be sure that I follow exactly what was done.

L179-180, Clarification: Here, the phasing “... for loci containing variable sites...” implies that in other cases monomorphic loci were included in the analysis. However, I would have thought that the latter would be excluded. Please clarify the treatment of invariable loci.
L185, Add words: “...loss of more variable loci [owing to them being subdivided into two or more less variable loci]” (or similar).

L195, Rephrase: “demographic history” is a little vague, as it could refer to changes in Ne over time vs. an estimate of the long-term (e.g. harmonic mean) of Ne, or perhaps something else.

L203-204: Clarification: priors for the two parameters are each reported as a single value rather than a range: were these really fixed, or free to vary (within bounds) around a measure of central tendency? If fixed, some additional rationale for user-specified values would be good.


RESULTS

L214-L215 (& Fig. 2a), Clarification: The y-axis of Fig 2a (average [?] # allele / locus), where values are bounded by ~1.2 and ~2.0, suggests that these calculations included monomorphic loci... otherwise values would be lower bounded by 2. If so, this needs to be clearly stated in the Methods section, as it is not clear how monomorphic loci were treated.

L216-216, Add words & reorganize: “The [difference in] number of alleles [among lineages] was more similar across stringent thresholds...” (& delete: “and this effect impacted relative values between lineages”).

L226-227 (& Fig. 3a): Do these (i.e., reported genetic distance estimates) refer to / represent p-distance or JC-corrected distance? The Methods section indicated that both were calculated and applied at the individual-level.


DISCUSSION

L274-277, Clarification: Is the “loss of the most divergent loci” due to them being reduced down to less polymorphic loci, or to their re-classification from polymorphic to monomorphic followed by omission from the dataset? In the case of Huang & Knowles et al., I think it is neither of these (but instead, due to omission of loci with “too much” missing data / genotypes).

L281 (& elsewhere), Clarification: I think there is an important difference between “loss of alleles” and “loss of loci”, in that the former relates to failure to identify / recognize the presence of some but not all alleles at a locus (often the genetically divergent ones), whereas “loss of loci” is more of an issue that relates to discarding an entire locus for one reason or another (e.g., too much missing data, misbehaving re: Mendelian expectations etc). Issues relating to over- & under splitting are a little different, in that alleles are not entirely lost, but are retained in an unusual way (i.e., separated into different loci). I wonder if new terminology is needed to more fully describe the issue at hand (and by extension, separate if from others).

L301-304, Rationale: “Our indirect measure of over-splitting may detect not just over-split loci, but also loci that are under-split in the assembly from the most liberal threshold but correctly separated in the assembly from the more stringent threshold”. Under what circumstances would the latter be true? Further explanation needed here.

L307: Is it really a “test” (cf. metric), given that no p-value is associated with it?

L317-319, Comment: Other pragmatic approaches to addressing the Question of "which alleles belong to a given locus" could be to (1) include a subset of pedigreed individuals in the panel of those to be genotyped for each species and then to assess Mendelian inheritance patterns at alternative similarity thresholds, and/or (2) for a group of individuals for which there are strong a priori reasons to believe are members of a single panmictic population, to assess deviations from HWE (and LE) under alternative similarity thresholds.

L330: “population history inferences” overly vague.


TABLES & FIGURES

Table 1 caption: define “SD”, and expand to define “samples per locus”

·

Basic reporting

Line 493 “assemhly” should probably read "assembly".

Lines 521-524: It is not clear to me what “Samples per Locus” is in Table 1. Average number or reads per locus?

Lines 570-572"Putative over-split loci (connected by dashed lines) were more frequent in datasets assembled at stringent similarity thresholds, whereas loci containing undersplit reads (solid lines) occurred at low frequency across all similarity thresholds.” I can only see one solid line (purple) in Fig 2b. Are they overlapping or do I misunderstand something?

Experimental design

No comment

Validity of the findings

Lines 521-524: Why is the number of loci at its lowest at 99% similarity for Xenops in Table 1?

Additional comments

This is a well written manuscript covering a very important topic, namely the similarity threshold used to reconstruct orthologous loci from short read sequencing and its effect on various parameters used in populations genetic studies.

The manuscript clearly describes these effects and I'm perfectly happy with as it is. However, as a plant systematist focusing on hybrids and polyploids - and as several of the study taxa hybridize - I would appreciate a discussion on that too. You write that undersplitting is minimal with low thresholds, but what happens if you happen to have young hybrids (with high levels of heterozygosity) or allopolyploids (with fixed heterozygosity) in your dataset?

This might be a bit outside the scope of the study so feel free to ignore this, but you mention that Stacks couldn't handle thresholds below 93%. However, several published studies using Pyrad use thresholds as low as 85% - would undersplitting become a problem at that level?

---

## Round 0.2 · accepted · Accept

In Fig. S3 you may want to provide the Greek letter theta instead of "Theta" in the axis label, but I will let you work with the PeerJ staff if you want to fix that (optional).